# Influence of Cutting Dimensions and Cooking Methods on the Nutritional Composition and Sensory Attributes of Zucchini (*Cucurbita pepo* L.)

**DOI:** 10.3390/foods14183213

**Published:** 2025-09-16

**Authors:** Adela Abellán, Pablo Gómez, Alba Villegas, Laura Buendía-Moreno, Luis Tejada

**Affiliations:** Faculty of Pharmacy and Nutrition, Universidad Católica de Murcia UCAM, Campus de los Jerónimos, 30107 Murcia, Spain; pgomez2@ucam.edu (P.G.); ltejada@ucam.edu (L.T.)

**Keywords:** *Cucurbita pepo* L., cutting size, culinary treatment, nutritional composition, bioactive compounds, sensorial

## Abstract

The present work investigated how cube size (10 × 10 × 10 mm and 20 × 20 × 20 mm) and culinary technique (stir-frying, steaming, with raw zucchini as control) influence the nutritional profile, bioactive components, texture, and sensory properties of zucchini (*Cucurbita pepo* L.). Parameters assessed included moisture, dry matter, ash, protein, fat, antioxidant capacity (AC), total phenolic content (TPC), texture profile, and consumer acceptance. Cutting size significantly affected dry matter and ash contents, with larger cubes showing higher values. While TPC remained stable across sizes and methods, AC varied with both factors, reflecting contributions from non-phenolic antioxidants and retention differences. Hardness was unaffected by cutting size, but raw larger cubes had higher adhesiveness; cooking markedly reduced both parameters. Stir-frying increased dry matter, ash, protein, fat, and AC, partly due to incorporation of oil-derived antioxidants, whereas steaming preserved visual attributes and produced the highest sensory scores for appearance, colour, and texture. Smaller cubes were generally preferred in sensory evaluation. In summary, the culinary technique exerted a stronger effect than cube size: stir-frying boosted specific nutritional parameters, whereas steaming was more effective at maintaining sensory quality.

## 1. Introduction

Recent modifications in eating habits, largely associated with the global increase in metabolic diseases such as cardiovascular disorders and diabetes, have reinforced the importance of fruit and vegetable consumption. Fruits and vegetables are well known for their low caloric density and high nutritional quality, providing fiber, vitamins, minerals, and numerous bioactive constituents, including carotenoids, phenolic compounds, glucosinolates, and antioxidants [1,2]. Such compounds play a key role in mitigating oxidative stress and reducing the risk of degenerative diseases.

Zucchini (*Cucurbita pepo* L.) is a widely consumed low-calorie vegetable whose main component is water, followed by carbohydrates, with minor contributions from protein and fat. It is a good source of folates and vitamin C, as well as potassium, magnesium, phosphorus, and iron [3]. Despite its nutritional importance and culinary versatility, the combined effects of cut size and cooking method on its chemical and sensory properties remain largely unexplored.

Although some studies have assessed how cutting methods influence the nutritional and sensory characteristics of vegetables such as onion, carrot, cabbage, and papaya, no such work has yet focused specifically on zucchini. For instance, researchers founded that slicing onions increased sugar content while grated onions were preferred for aroma and appearance [4]. Similarly, cutting carrots into dices better preserved their sensory and nutritional qualities compared to slicing. It was demonstrated that slicing enhanced the quality index of fresh-cut papaya during refrigerated storage [5]. Meanwhile, other studies have documented discoloration and microbial spoilage as key factors limiting the shelf life of minimally processed carrots [6,7].

Thermal processing also induces significant physical, chemical, and sensory modifications in vegetables. Nutritionally, cooking alters both the levels and the bioavailability of essential nutrients and bioactive compounds [8]. The type of heat treatment applied must be tailored to the characteristics of each food to minimize nutrient loss and optimize texture and flavor [9,10]. Cooking promotes moisture loss, protein coagulation, fiber softening, and dissolution of chemical compound processes that also affect organoleptic attributes.

Recent studies highlight that microwave cooking and steaming preserve nutrients more effectively than conventional boiling. The retention of minerals and limited polyphenol loss (8–56%) in zucchini were observed when cooked by microwave or steam, compared with higher losses from boiling [11]. Complementing this, a 2024 review demonstrated decreasing bioactive compound retention in the sequence microwave > steam > boiling. Additionally, structural transformations of carbohydrates during cooking can alter fiber profiles: while soluble fibers may solubilize and decrease, thermal processing can enhance measured fiber via formation of fiber–protein complexes.

Moreover, studies show that heating vegetables such as spinach, cauliflower, and cabbage alters their antioxidant properties and sensory profiles [12,13]. Understanding physicochemical changes during cooking is essential to predict nutritional losses and gains, enhance digestibility, and reduce microbial risk while also promoting the formation of desirable compounds like antioxidants and flavor precursors [14].

Importantly, food composition tables often reflect raw product values and overlook compositional changes induced by culinary treatments. There is a clear lack of studies assessing how different cooking methods impact the nutritional characteristics of fresh zucchini, especially in relation to cut size. Accordingly, the objective of this work was to investigate how two different cube sizes (10 × 10 × 10 mm and 20 × 20 × 20 mm) and two culinary methods (stir-frying and steaming) influence the nutritional composition, bioactive compounds, texture, and sensory properties of zucchini, taking raw diced samples as the reference.

## 2. Materials and Methods

### 2.1. Sample Preparation

Commercially mature zucchini was sourced from a frozen vegetable supplier in Murcia. To control for environmental variability, all fruits were collected during April–May from plants cultivated in the same greenhouse. A total of three 60-kg lots were obtained; each was divided equally, producing 30 kg cut into 10 × 10 × 10 mm cubes and 30 kg cut into 20 × 20 × 20 mm cubes. From each of these six groups (three small-cube and three large-cube batches), 1000 g portions were taken for analysis of raw, stir-fried, and steamed samples.

### 2.2. Cooking Methods

Two thermal processing methods were applied: stir-frying and steam cooking.

For stir-frying, 1000 g of zucchini cubes (either 10 × 10 × 10 mm or 20 × 20 × 20 mm) were prepared using 64 g of extra virgin olive oil at 250 °C, in line with customary Mediterranean cooking practices. The procedure lasted 8 min for the smaller cubes and 11 min for the larger cubes. For steaming, 1000 g from each batch were processed in a Rational oven (Nisbets Plc., Fourth Way, Avonmouth, Bristol, UK) operating at 100 °C and full humidity. Steaming times were 13 min for 10 × 10 × 10 mm cubes and 17 min for 20 × 20 × 20 mm cubes. The vegetables were arranged on perforated gastronorm trays to allow drainage of cooking water without direct contact with the samples. Following immediate measurements of moisture content, texture, and sensory attributes, the cooked zucchini were homogenized in a Thermomix TM5 (Vorwerk, Wuppertal, Germany), transferred into sterile bottles, and stored at approximately 4 °C until further analysis.

### 2.3. Physicochemical Characterisation

Moisture, ash content, proteins, and fat were determined in all the samples of raw, stir-fried, and steamed zucchini (10 × 10 × 10 and 20 × 20 × 20 mm. Moisture was quantified following AOAC 2020 guidelines [15]. Ash content was obtained by incinerating the material at 550 °C [15]. Protein and fat were assessed using the Kjeldahl and Soxhlet methods, respectively [15].

### 2.4. Analysis of Bioactive Compounds

#### 2.4.1. Extraction of Sample

Extraction was carried out according to the method of Ferracane et al. (2008), with some adjustments [16]. Ten grams of zucchini were mixed with 40 mL of 60% ethanol and centrifuged at 1000 rpm for 5 min at 25 °C to obtain the precipitate. This step was repeated four times. The combined supernatants were vacuum-dried below 30 °C, and the residue was re-dissolved by ultrasonic stirring to a final volume of 20 mL. Extracts were stored at −18 °C until further analysis.

#### 2.4.2. Phenolic Compound Determination

The total phenolic content was determined with the Folin–Ciocalteu assay [17]. A mixture of 40 µL extract, 0.5 mL Folin–Ciocalteu reagent, and 2 mL of 20% Na_2_CO_3_ was prepared, then diluted with distilled water to 10 mL. Samples were kept in the dark at 25 °C for two hours. Absorbance was read at 765 nm using a UV-2550 spectrophotometer (Shimadzy, Kyoto, Japan). Results were calculated from a calibration curve and expressed as µg gallic acid equivalents per gram of dry extract (GAE, mg gallic acid/g dry extract).

#### 2.4.3. Antioxidant Activity Determination

Antioxidant activity was assessed by the DPPH (2,2-diphenyl-1-picrylhydrazyl) free radical scavenging method [18], with modifications. A 500 µL aliquot of zucchini methanolic extract was combined with 500 µL ethanol and 125 µL of 0.02% DPPH solution. The mixture was vortexed and kept in the dark for 60 min. Absorbance was recorded at 517 nm with a UV-2550 spectrophotometer (Shimadzy, Kyoto, Japan), using methanol without DPPH as the blank. Inhibition percentages of the DPPH radical were then calculated according to Equation (1).% inhibition of DPPH = ((Abs CONTROL − Abs SAMPLE)/(Abs CONTROL)) × 100(1)
where Abs CONTROL is the absorbance of DPPH with methanol, and Abs SAMPLE is the absorbance of the DPPH with the zucchini methanolic extract. The antioxidant activity was calculated as ascorbic acid equivalent antioxidant capacity (AAE, mg ascorbic acid/g dry extract).

### 2.5. Instrumental Texture Analysis

Texture measurements were carried out using a Texture Analyser (TAXT2, Stable Micro System, Godalming, UK) set with a force of 0.05 N. A shearing/cutting test was performed with the HDP/KS5 probe in combination with a Kramer Shear Cell fitted with five blades. The test was conducted over a 50 mm distance at a speed of 1 mm/s. Hardness (g) and adhesiveness (mJ) values were obtained.

### 2.6. Sensory Analysis

The sensory analysis was conducted in line with international standards. The tests were developed in a standard room equipped with 10 individual tasting areas [19]. In the test 25 non-trained panellists participated.

This consumer acceptance test was conducted with adult untrained panelists who tasted zucchini cooked under different treatments and rated appearance, odour, taste, texture, hardness, adhesiveness and overall liking using a structured five-point hedonic scale, 1 being I dislike it a lot and 5 being I like it a lot.

Participants provided verbal consent and were informed that the food served was safe and commonly consumed. According to national and institutional guidelines, this study was considered low-risk sensory research and an ethical review was not required.

Each panelist received a questionnaire along with the cooked samples, which were identified with two random codes. Samples were served at room temperature on glass plates, and mineral water was provided to cleanse the palate between tastings [20].

### 2.7. Statistical Analysis

All analyses, apart from the sensory evaluation, were carried out in triplicate and results reported as mean ± standard deviation. A two-way ANOVA combined with Tukey’s test was applied to assess the influence of cube size (10 × 10 × 10 and 20 × 20 × 20 mm) and cooking method on nutritional composition, bioactive compounds, and sensory outcomes of zucchini. Differences were considered statistically significant at *p* < 0.05. Statistical processing was performed with SPSS software version 21.0 (IBM Corporation, Armonk, NY, USA).

## 3. Results

### 3.1. Nutritional and Chemical Profile

Table 1 shows the results obtained for the effect of cutting size and culinary treatment on the nutritional composition of diced zucchini. The cutting size significantly affected the dry extract and ash content, higher values were seen in zucchini dice of size 20 × 20 × 20 mm than the 10 × 10 × 10 mm dice. Although no significant differences were observed in the other nutritional parameters as a function of cutting size (*p* > 0.05), fat content analysis revealed a significant interaction between cutting size and cooking method. Specifically, stir-fried zucchini cubes of 20 × 20 × 20 mm contained higher fat levels than those cut to 10 × 10 × 10 mm. Furthermore, stir-frying produced significant increases (*p* < 0.05) in dry matter, ash, protein, and fat contents, whereas steaming did not induce significant changes in any of the nutritional variables when compared with raw zucchini.

### 3.2. Antioxidant Properties

Table 2 shows the results of phenol and antioxidant activity of the different cut sizes of and prepared zucchini. The phenol content was not significantly affected (*p* > 0.05) by the cut size, but by cooking, in diced zucchini (20 × 20 × 20 mm) stir-fried and steamed, an increase in phenolic compounds content (0.16 µg AEG/g for both techniques) was observed, compared to the diced raw zucchini (0.14 µg AEG/g). No significant differences were observed in the total phenol content between fresh and cooked 10 × 10 × 10 mm diced zucchini. The antioxidant activity increased significantly after the courgettes were cooked, mainly when they were stir-fried. Higher values were also observed in the cut zucchini (20 × 20 × 20 mm), steamed and stir-fried, (19.66 and 39.77 mg AA/dry weight, respectively) in relation to fresh diced (13.40 mg AA/dry weight). Thus, the cut size did not significantly affect antioxidant activity in fresh zucchini.

### 3.3. Instrumental Texture Analysis

Table 3 shows the effect of cutting size and cooking method on the texture parameters of diced zucchini. Cutting size had no significant effect on hardness (*p* > 0.05), but it did influence adhesiveness, which was significantly higher in the 20 × 20 × 20 mm cubes; however, this difference was only evident in raw samples. Culinary treatment reduced both hardness and adhesiveness compared with raw zucchini. Among cooking methods, steaming produced the softest texture (10,756 g), followed by stir-frying (20,370 g), while raw zucchini exhibited the highest hardness (49,500 g). Stir-fried samples retained considerably more firmness than steamed ones. Adhesiveness did not differ significantly between steaming (63.83 mJ) and stir-frying (48.02 mJ), but both values were markedly lower than in raw zucchini (1016.68 mJ). These results indicate that cooking substantially reduces both firmness and stickiness, regardless of cutting size, with steaming having the greatest softening effect.

### 3.4. Sensorial Characteristics

Table 4 summarises the influence of cutting size and cooking method on the sensory attributes of diced zucchini. Cutting size and culinary treatment significantly affected the appearance, colour, and texture characteristics of cooked zucchini, receiving 10 × 10 × 10 mm cubes higher ratings than 20 × 20 × 20 mm cubes. In contrast, no significant differences were detected in odour, taste, or overall acceptance between sizes (*p* > 0.05).

Cooking method also affected several sensory parameters. Steamed zucchini achieved the highest scores for appearance, colour, odour, and texture, often with significant differences compared with stir-fried samples. For example, 10 mm cubes scored 3.98 for appearance and 3.82 for texture when steamed, compared with 3.46 and 3.38, respectively, when stir-fried. These trends suggest that steaming better preserves visual quality and texture perception, possibly due to reduced surface browning and softer texture development, while stir-frying may induce colour darkening and firmer texture that slightly reduce consumer preference. Despite these differences, general acceptance did not vary significantly among treatments, indicating that both cooking methods produced zucchini with acceptable sensory quality.

## 4. Discussion

In vegetables, when plant tissue is damaged by cutting, the internal tissues are exposed and the evaporation rate of the water increases, thus the dry extract, which justifies the results obtained for this parameter [21].

The influence of cutting size on moisture loss can be explained by differences in the surface-to-volume ratio (S/V) of the zucchini cubes. Smaller pieces (10 × 10 × 10 mm) present approximately twice the S/V of larger cubes (20 × 20 × 20 mm), which increases the exposed surface area and reduces the diffusion path length for water migration. This geometric configuration facilitates faster moisture evaporation and greater final water loss, a phenomenon consistently reported in drying and cooking studies for fruits and vegetables [22,23,24,25]. Comparable trends have been observed in zucchini, where reducing slice thickness significantly accelerates dehydration rates and decreases final moisture content under similar processing conditions [26].

Researchers analyzed the influence of the cut (grated, cubes, and julienne) on the physiological behaviour of vegetables, including carrots [27]. In the study, when the surface/volume ratio of the vegetable was higher, there was a greater loss of water, so the grated carrot presented losses higher than the cubes and julienne. The influence of the cut on the quality of onions has also been examined [28], observing that the sliced onion presented higher quality for a longer time than the chopped and grated onions.

The higher ash content in larger zucchini cubes (20 × 20 × 20 mm) is mainly due to greater water loss combined with lower mineral leaching. Because ash is expressed on a fresh-weight basis, losing more water while retaining most minerals results in a higher concentration. In contrast, smaller cubes have more surface area, which can promote the loss of water-soluble minerals during cooking [22].

Exposure to high cooking temperatures, such as during stir-frying, leads to moisture loss and consequently raises dry matter content, likely associated with tissue disruption in the vegetable that promotes further water release [29].

Researchers observed that moisture decreases as the temperature during cooking is higher [30]. Thus, justifying that the stir-fried zucchini has a lower moisture and a higher dry extract content when compared to steam cooking. The high ash content in stir-fried zucchini, compared to the other treatments, may be because no water is used when stir-frying, and therefore implies less loss of water-soluble components. The elevated fat content in stir-fried diced zucchini results from water evaporation at the surface layers, which allows oil to penetrate and act as the heat transfer medium [9,31,32].

The difference in fat content between 20 mm and 10 mm stir-fried zucchini cubes can be understood as the outcome of two related processes: water loss and oil dynamics. Larger cubes underwent greater net dehydration during cooking, and because fat values are expressed relative to fresh weight, this moisture reduction increases the apparent fat content. Their lower surface-to-volume ratio may also limit rapid surface crusting and facilitate deeper oil penetration before the surface dries. Smaller cubes, despite cooking through more quickly, present more cut surfaces where oil can drain back into the cooking medium, and rapid dehydration may trap less oil inside the tissue. Similar patterns have been reported in frying studies where geometry influences both the rate of moisture migration and the retention of surface and internal oil [22,33,34,35].

During food cooking, transformations take place that affect the composition and nutritional value of the food, to improve its sensory characteristics. Stir-frying causes the water to evaporate from the food, and the absorbed oil partially replaces the released water, causing an increase in its caloric value.

In this study, zucchini cutting size influenced antioxidant capacity more markedly than total phenolic content (TPC). While TPC remained relatively stable across sizes within each cooking method, antioxidant capacity (AC) varied significantly, suggesting the contribution of compounds beyond phenolics. Phenolic compounds are soluble in water, so cooking techniques that involve water, such as steam cooking, lead to phenolic losses from the food [36]; hence the results obtained in this study. However, the increase in the antioxidant activity of zucchini during stir-frying is due to using extra virgin olive oil, since it provides the food with its own antioxidants such as tocopherol, squalene, and avenasterol, enriching the vegetable with these vitamins, and hence the high values compared to steam cooking [9].

Non-phenolic antioxidants such as vitamin C, carotenoids, and lipophilic antioxidants from extra virgin olive oil (in the case of stir-frying) can substantially affect AC [37,38,39]. During steaming, the larger cubes likely experienced lower leaching of water-soluble antioxidants due to their lower surface-to-volume ratio, leading to higher AC values compared with smaller cubes. In stir-frying, the increase in AC, particularly in larger cubes, may also be linked to the incorporation of tocopherols, squalene, and sterols from the olive oil, which enrich the vegetable with additional antioxidant compounds [40]. These findings indicate that AC in cooked zucchini is the result of a complex interplay between intrinsic compounds, their stability during cooking, and the potential enrichment from the cooking medium, rather than TPC alone.

Texture is a key determinant of consumer acceptance in fresh and cooked zucchini. In this study, cutting size did not significantly influence the hardness of raw zucchini cubes, suggesting that firmness in fresh tissue is governed more by intrinsic cell structure than by geometry. However, adhesiveness was significantly higher in larger raw cubes, likely due to their lower surface-to-volume ratio, which reduces the rate of surface moisture loss after cutting and retains more soluble solids at the surface, enhancing stickiness during texture analysis. Smaller cubes, with proportionally more exposed surface, tend to lose free moisture faster and release more cellular fluids during handling, which may diminish adhesiveness [41].

Cooking markedly reduced both hardness and adhesiveness compared with raw samples, reflecting the combined effects of pectin and hemicellulose degradation in cell walls and loss of turgor pressure. Steaming produced the greatest softening, while stir-frying preserved more firmness, possibly due to rapid surface dehydration that limits internal water migration. Furthermore, adhesiveness is strongly influenced by the distribution of water and soluble polysaccharides such as pectins and mucilaginous compounds. Larger pieces retain more intracellular water and soluble matrix components, increasing stickiness during compression, while smaller pieces exhibit less retention of these components [41,42,43,44].

In this study, the hardness of diced raw zucchini was not affected by cutting, also seen in a study conducted on cut papaya (dices and slices) stored at different refrigeration temperatures [45]. Temperature influences the texture of vegetables, since the higher the storage temperature, the greater the loss of firmness. Such an effect has been reported for other fresh cut products such as peaches, pears, and bell peppers [46]. Losing firmness may be due to the hydrolysis of the components of the cell wall, consisting mainly of pectins, hemicelluloses, and cellulose polysaccharide polymers, due to the friction produced by cutting. Therefore, the rapid deterioration of the freshly cut vegetable may be due to the enhanced activity of the enzymes that hydrolyse these cell wall components and consequently to an accelerated senescence of the products [47]. In this work, cooking treatment altered the hardness and adhesiveness of diced zucchini when compared with raw samples. These textural modifications are linked to heat-induced degradation of pectin and hemicelluloses, which are structural components of plant cell walls [48].

The sensory evaluation revealed that smaller zucchini cubes (10 × 10 × 10 mm) were consistently rated higher in appearance, colour, and texture than larger cubes, regardless of cooking method. This preference may be linked to the more uniform cooking and softer texture achieved in smaller pieces, as well as their visually appealing, more regular geometry. Similar consumer responses to smaller cut sizes have been reported in other minimally processed vegetables, where uniform appearance and tenderness contribute to higher visual and textural scores [43,49].

Cooking method also influenced sensory perception. Steamed zucchini generally received the highest ratings for appearance, colour, odour, and texture, which may be attributed to the preservation of natural pigments and moisture, and the absence of browning reactions typically induced by high-temperature methods. Steaming limits oxidative degradation of chlorophyll and carotenoids, maintaining greener hues and brighter appearance, while also producing a softer texture that many consumers associate with tenderness and freshness [37,38].

In contrast, stir-fried samples tended to score lower for appearance and colour, likely due to slight surface browning and oil absorption, which can darken the product and modify its surface texture. While these changes are not necessarily negative from a culinary standpoint, they may slightly reduce perceived freshness and uniformity in a controlled sensory panel. Nevertheless, overall acceptance scores did not differ significantly between treatments, indicating that both steaming and stir-frying produced zucchini of acceptable sensory quality, even if certain attributes were rated higher for steaming.

The visual properties of freshly cut zucchini are an important parameter that the consumer considers when evaluating its quality. The effect of the cut had a significant effect on consumer satisfaction, the smaller cut being the best valued. It was observed that the rate of deterioration in cut carrots is influenced by the size of the cut sections, therefore the zucchini diced with the largest surface area were the worst valued [50]. However, applying heat to prepare a cooked dish is conducted to modify the sensory characteristics of the food and improve its palatability. The temperatures used contribute to the inactivation of the microbial load, to eliminating enzymes, and to the decrease in the water activity, depending on the culinary technique applied. The sudden contact of the diced zucchini with the oil coagulates its proteins, induces the caramelisation of the starch, and favours Maillard reactions. The water content of the inner part of the zucchini is reduced and the outer part of the rind is dehydrated. Therefore, the stir-fried dice zucchini was the worst in terms of sensory attributes valued by the consumer. This can also be due to the oil reacting with oxygen in the air, creating oxidation products and providing the food with unwanted flavours. However, the cooked diced zucchini presented a more pleasant and edible texture, since the culinary technique used provided these qualities to the food.

## 5. Conclusions

Cutting size influenced some zucchini properties, with larger cubes showing higher dry matter and ash contents and smaller cubes receiving higher sensory ratings for appearance, colour, and texture. Cooking method had a stronger effect: stir-frying enhanced certain nutritional parameters and antioxidant capacity but increased caloric value, while steaming better preserved visual quality and texture. Both methods yielded acceptable sensory quality, with the choice depending on whether nutritional enrichment or sensory preservation is prioritised.

## Figures and Tables

**Table 1 foods-14-03213-t001:** Changes in nutrition composition of cutting size zucchini during cooking (g/100 g of zucchini).

Culinary Treatment	Cutting Size	Dry Extract	Ash	Protein	Fat
Raw	10 × 10 × 10	4.66 ± 0.45 ^d^	0.32 ± 0.11 ^b^	1.51 ± 0.07 ^c^	0.08 ± 0.00 ^c^
	20 × 20 × 20	5.23 ± 0.08 ^cd^	0.62 ± 0.02 ^cd^	1.52 ± 0.07 ^c^	0.07 ± 0.01 ^c^
Steam	10 × 10 × 10	4.96 ± 0.54 ^cd^	0.55 ± 0.01 ^d^	1.34 ± 0.03 ^c^	0.06 ± 0.06 ^c^
	20 × 20 × 20	5.53 ± 0.16 ^c^	0.65 ± 0.00 ^cd^	1.16 ± 0.08 ^bc^	0.04 ± 0.01 ^c^
Stir-frying	10 × 10 × 10	8.19 ± 0.63 ^b^	0.82 ± 0.15 ^c^	2.08 ± 0.17 ^a^	0.44 ± 0.16 ^b^
	20 × 20 × 20	10.18 ± 0.73 ^a^	1.17 ± 0.06 ^a^	2.01 ± 0.25 ^ab^	0.95 ± 0.06 ^a^
**Results of ANOVA**					
*p*-value	Cutting size	0.000001	0.000608	0.423675	0.090558
	Culinary treatment	0.000000	0.000001	0.000386	0.000000
	Interaction	0.001567	0.259270	0.306820	0.000174

^a,b,c,d^ Different letters in the same column indicate a significant difference (*p* ≤ 0.05) between the samples. Values are mean ± SD (n = 3).

**Table 2 foods-14-03213-t002:** Changes to the bioactive compounds of zucchini according to cutting size during cooking.

Culinary Treatment	Cutting Size	Total Phenolic (µg AEG/Dry Weight)	Antioxidant Capacity (mg AA/Dry Weight)
Raw	10 × 10 × 10	0.15 ± 0.01 ^ab^	13.96 ± 0.70 ^d^
	20 × 20 × 20	0.14 ± 0.00 ^b^	13.40 ± 0.74 ^d^
Steam	10 × 10 × 10	0.15 ± 0.01 ^ab^	16.20 ± 0.23 ^cd^
	20 × 20 × 20	0.16 ± 0.0 ^a^	19.66 ± 0.63 ^bc^
Stir-frying	10 × 10 × 10	0.14 ± 0.01 ^b^	21.76 ± 2.68 ^b^
	20 × 20 × 20	0.16 ± 0.0 ^a^	39.77 ± 3.03 ^a^
**Results of ANOVA**			
*p*-value	Cutting size	0.061648	0.000028
	Culinary treatment	0.437487	0.000000
	Interaction	0.036197	0.000019

^a,b,c,d^ Different letters in the same column indicate a significant difference (*p* ≤ 0.05) between the samples. Values are mean ± SD (n = 3). µg AEG/dry weight: microgram of gallic acid equivalent per gram of dry weight. mg AA/dry weight: milligram of ascorbic acid equivalent per gram of dry weight.

**Table 3 foods-14-03213-t003:** Changes in the texture of cutting size zucchini during cooking.

Culinary Treatment	Cutting Size	Hardness (g)	Adhesiveness (mJ)
Raw	10 × 10 × 10	51,000 ± 2880 ^a^	868.37 ± 46.73 ^b^
	20 × 20 × 20	49,500 ± 2270 ^a^	1016.68 ± 35.44 ^a^
Steam	10 × 10 × 10	12,015 ± 504 ^c^	41.87 ± 4.39 ^c^
	20 × 20 × 20	10,756 ± 1946 ^c^	63.83 ± 9.92 ^c^
Stir-frying	10 × 10 × 10	18,896 ± 1484 ^b^	39.03 ± 2.66 ^c^
	20 × 20 × 20	20,370 ± 1677 ^b^	48.02 ± 3.97 ^c^
**Results of ANOVA**			
*p*-value	Cutting size	0.794850	0.007044
	Culinary treatment	0.000000	0.000000
	Interaction	0.704947	0.016951

^a–c^ Different letters in the same column indicate a significant difference (*p* ≤ 0.05) between the samples. Values are mean ± SD (n = 3); g: grams; mJ: millijoules.

**Table 4 foods-14-03213-t004:** Changes in the sensory attributes of zucchini according to cutting size during cooking.

Culinary Treatment	Cutting Size	Appearance	Colour	Odour	Taste	General Texture	General Acceptation
Steam	10 × 10 × 10	3.98 ± 0.13 ^a^	4.06 ± 0.11 ^a^	3.44 ± 0.12 ^a^	3.24 ± 0.16 ^a^	3.82 ± 0.13 ^a^	3.50 ± 0.12 ^a^
	20 × 20 × 20	3.20 ± 0.18 ^b^	3.35 ± 0.16 ^c^	3.28 ± 0.14 ^ab^	3.33 ± 0.17 ^a^	3.28 ± 0.18 ^b^	3.28 ± 0.17 ^a^
Stir-frying	10 × 10 × 10	3.46 ± 0.18 ^b^	3.56 ± 0.17 ^c^	3.28 ± 0.16 ^ab^	3.18 ± 0.17 ^a^	3.38 ± 0.15 ^b^	3.34 ± 0.14 ^a^
	20 × 20 × 20	3.00 ± 0.19 ^b^	2.80 ± 0.18 ^b^	3.04 ± 0.15 ^b^	3.11 ± 0.17 ^a^	3.04 ± 0.14 ^b^	3.28 ± 0.15 ^a^
**Results of ANOVA**							
*p*-value	Cutting size	0.000330	0.000009	0.167073	0.964733	0.004166	0.343277
	Culinary treatment	0.036739	0.001709	0.161550	0.407146	0.025333	0.580765
	Interaction	0.341602	0.943694	0.780833	0.637909	0.505746	0.580765

^a–c^ Different letters in the same column indicate a significant difference (*p* ≤ 0.05) between the samples. Values are mean ± SD.

## Data Availability

The original contributions presented in this study are included in the article. Further inquiries can be directed to the corresponding author.

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
