# Peer review of "Influence of Cutting Dimensions and Cooking Methods on the Nutritional Composition and Sensory Attributes of Zucchini (Cucurbita pepo L.)"

_foods, 2025, doi:10.3390/foods14183213_

Round 1
Reviewer 1 Report
Comments and Suggestions for Authors
Manuscript title - Nutritional and sensorial characteristics of zucchini (Cucurbita pepo L.) as affected by cutting size and the culinary treatment
Manuscript ID – foods-3808453
The above-entitled manuscript is the evaluation of zucchini for different cooking condition at different sizes as against the raw samples to see its effect on proximate composition, phenolic compounds, antioxidant activities, sensory characteristics (texture, hardness, and adhesiveness) as well as hedonic sensory analysis. It is an interesting study area to evaluate different cooking methods for its physicochemical, nutritional and phytochemical comparison so that the study results could be used for selecting cooking methods and sample size for better nutritional values of zucchini. While reviewing this manuscript, I found that the authors did a couple of analysis and their discussion from the available literatures to come to the stated conclusions. However, I found many redundant serious issues, as below-listed pointwise, should be first addressed before processing further:
Abstract: Cubes should be three measurements, not two as mentioned here as 10 × 10 mm or 20 × 20 mm. Please correct it accordingly. The same issue is prevailed throughout the manuscript.
Introduction (Line 70 - 71): Three cooking methods (raw, stir-fried, and steamed) – raw is not a cooking method. Please correct it accordingly.
Sample preparation: For stir-frying, 64 g of extra virgin oil was used for 1000 g of sample at 250 oC for 8 min (10 mm cubes) or for 11 min (20 mm cubes) – What was the basis of selecting these values, especially 64 g of oil, 8 min for 10 mm cubes and 11 min for 20 mm cubes? Please briefly explain here.
The use of P < 0.05 should be proper – the authors should understand when to use P < 0.05 and when to use P > 0.05. Please amend accordingly. Also, P should be italic in P < 0.05.
Table 1: For raw sample (10 and 20 mm cubes), ash content was almost double (0.32 and 0.62 g/100 g, respectively) - Why so much different ash content just because of the sample size? Same question for the stir-fried sample.
Table 1: Fat contents for stir-fried samples (10 and 20 mm cubes) were 0.44 and 0.95 g/100 g, respectively. Please explain the possible reason for such difference in values just because of the sample size.
Table 2: For steamed and stir-fried samples with different sizes, TPC values were the same but antioxidant capacity was different. What could be the possible reasons? Please explain with appropriate references.
Table 3: Adhesiveness value for raw sample – 868 mJ for 10 mm cubes and 1016 mJ for 20 mm cubes. These samples were simply raw samples (without any cooking) – why such different in adhesiveness value just because of size difference?
Line 219: “…. larger dice zucchini, due to the higher surface area/volume ratio….” – this statement is not correct from a basic science logic. Please amend accordingly.
Conclusions: “Moreover, consumers gave better sensory valued to the smaller cubes zucchini. Therefore, the cut size used in this study does not affect the nutritional and sensory properties of zucchini.” These two sentences contradict each other – please amend accordingly.
Line 286 – 287 “Therefore, the stir-fried dice zucchini were the worst sensory valued by the consumer.” Vs. line 300 – 301 “Therefore, it is considered that the most suitable method for cooking zucchini is stir-frying” – these two sentences contradict each other. Please amend accordingly.
English language and grammar need thorough revision to better understand the messages from this manuscript. A thorough correction from a professional manuscript writer is recommended.
The iThenticate report percent match of 39% is a bit higher; please reduce it to below 20%.
Comments on the Quality of English Language
English language and grammar need thorough revision to better understand the messages from this manuscript.
Author Response
We would like to thank the reviewers for their detailed and valuable comments, which have contributed significantly to improving the clarity and scientific quality of our manuscript.
We have rewritten most of the manuscript in order to address the reviewers’ suggestions, improve the English, and attempt to reduce the iThenticate report percent match. In addition to the revised version, we are submitting the responses to all three reviewers.
Below, we respond point by point, please, see the attachment.

Reviewer 2 Report
Comments and Suggestions for Authors
The article conducted a study with zucchini. The antioxidant activity, content of phenolic compounds, nutritional value and other properties of zucchini prepared in three different ways were evaluated. The size of the zucchini cubes and its influence on the previously mentioned parameters were also studied.
My comments and questions:
- Check out the title of the article, it shows a lot of plagiarism (Also other parts of the article).
- Affiliations font is too small.
- The abstract should be edited. Especially the last sentence.
- Lines 56-60: The information lacks a scientific source. Can you comment on why boiling results in less antioxidant compounds being lost than steaming or microwaving?
- Olive oil (stir-frying method) increased the amount of protein, fat, phenolic compounds, and antioxidant activity - due to the chemical compounds it contains. Do you think this preparation method is the most effective for preparing zucchini? Argue.
- Shouldn't Table 4 include the data for "raw" zucchini?
- The zucchini pieces were all the same size, but they were cut from different parts of the vegetable (middle, end, near the stem, etc.). Does the nutritional value and content of compounds vary from place to place? Does the cooking method make a difference (for different parts of the zucchini)?
- Is it effective to compare production methods when one of them uses a different raw material (olive oil), which has a significant impact on the nutritional value and content of compounds in the dish?
Reviewer 3 Report
Comments and Suggestions for Authors
Detailed comments are noted in the pdf file

Reviewer 4 Report
Comments and Suggestions for Authors
This study addresses an underexplored yet relevant topic: the combined impact of cutting size and culinary methods on zucchini’s nutritional, bioactive, and sensory properties. The experimental design is rigorous, methodology is well-documented, and statistical analyses are appropriate. The findings provide practical insights for food processing and dietary recommendations. However, some sections require clarification, and conclusions need refinement to align with results.
1.The abstract and conclusions state "cut size does not affect nutritional properties," yet Table 1 shows significant differences in dry extract and ash content between 10×10 mm and 20×20 mm cubes (P<0.05). Similarly, Section 3.3 notes cut size significantly affects adhesiveness in raw zucchini. Revise conclusions to acknowledge these impacts.
2.While stir-frying enhances antioxidant activity (Table 2), it also increases fat content by >10× (Table 1), raising caloric density. The conclusion recommends stir-frying as "most suitable" without addressing trade-offs (e.g., reduced sensory scores in Table 4). Discuss nutritional trade-offs explicitly and temper the recommendation.
3.Phenolic compounds are claimed "unaffected by cut size or cooking" (Abstract), yet Table 2 shows interactions (P=0.036) and significant increases in 20×20 mm cooked samples. Clarify this discrepancy and discuss possible mechanisms (e.g., surface-area-dependent leaching or oil infusion during stir-frying).
4.Untrained panelists (n=25) may introduce bias. Justify this approach (e.g., cost constraints) or cite validation for consumer tests. Additionally, Table 4’s ANOVA results for sensory attributes appear misaligned (e.g., "Smell" P=0.167 listed as "0.167070.96473"). Verify statistical reporting.
5.Cooking times differ for cutting sizes (stir-frying: 8 vs. 11 min; steaming: 13 vs. 17 min), potentially confounding size effects with thermal exposure. Acknowledge this limitation or justify time adjustments based on preliminary tests.
6.Section 2.1: Specify the rationale for 250°C stir-frying temperature (industry standard? smoke point of olive oil?).
7.Table 1: Use consistent significant notation (e.g., superscript letters a,b,c,d vs. Table 3’s asterisks).
8.Discussion
Bioactive Compounds: Contrast the stability of zucchini phenolics with other vegetables (e.g., carrots [28]) to highlight crop-specific behaviors.
Round 2
Reviewer 1 Report
Comments and Suggestions for Authors
The authors have not properly answered the comments no. 5, 6, 7 and 8. I suggest to address these issues properly based on scientific literatures or common sense. Moreover, iThenticate match percent is still an issue. Thank you!
Author Response
We sincerely appreciate your careful re-review of our manuscript
Please find the detailed responses in the PDGF attached.
A new version of the manuscript has been uploaded to the platform.

Reviewer 3 Report
Comments and Suggestions for Authors
The quality of work is improved, so it can be recommended for acceptance
Author Response
We are truly grateful to the reviewer for accepting the publication of our article and for their constructive contributions throughout the review process.
Reviewer 4 Report
Comments and Suggestions for Authors
Accept
Comments on the Quality of English LanguageAccept
Author Response

(The authors gave the same response as above.)
